# Investigation on Descriptive Epidemiology, Geographical Distribution, and Genotyping of *Echinococcus granulosus* s.l. in Bovine from Romania

**DOI:** 10.3390/vetsci9120685

**Published:** 2022-12-09

**Authors:** Gheorghe Dărăbuș, Amelia Bușe, Ion Oprescu, Sorin Morariu, Narcisa Mederle, Marius Ilie, Mirela Imre

**Affiliations:** Faculty of Veterinary Medicine, University of Life Sciences “King Mihai I” from Timișoara, Calea Aradului no. 119, 300645 Timişoara, Romania

**Keywords:** cattle, metacestodes, *E. granulosus*

## Abstract

**Simple Summary:**

During a period of two years (2020–2021), the study investigated the presence and geographical distribution of cystic echinococcosis, as well as the genetic characterization of *Echinococcus granulosus* s.l. in slaughtered cattle in Romania. From a total of 2693 examined cattle, 66 (2.45%) harbored cysts, with a dominant occurrence at the level of the lungs. Out of them, 6, 11, 41, and 8 cattle belonged to the 2–12, 13–16, 17–20, and >21 years age categories, respectively. The polymerase chain reaction (PCR) analysis of 10 randomly selected samples showed positive results for DNA fragments detection in 5 samples, which were identified as *Echinococcus granulosus sensu stricto* (s.s.). The study results indicate that the disease is still present in several of the investigated regions of Romania, highlighting a potential public health risk.

**Abstract:**

Bovine echinococcosis is a zoonotic disease with worldwide distribution, causing significant economic losses in the affected animals and important public health concerns. The aim of the current study was to investigate the presence and geographical distribution of cystic echinococcosis, and molecular characterization of *Echinococcus granulosus* s.l. from slaughtered cattle in Romania. In the period 2020–2021, a total of 2693 cattle (aged 2–21 years) slaughtered from the breeds Bruna de Maramureș, Bălțată Românească, and a mixed breed were examined to identify hydatid cysts. Cysts were identified in 66 cattle (2.45%). The predominant location of hydatid cysts was the lungs. Most cysts were non-fertile. By age categories, hydatidosis was reported in the age categories 2–12 years (6/2341; 0.25%), 13–16 years (11/244; 4.5%), 17–20 years (41/85; 48.23%), and over 21 years (8/23; 34.78%). Following the PCR analysis of 10 samples from the lungs (protoscoleces/proliferative membrane/hydatid liquid), DNA fragments were identified in 5 samples. Following sequencing, the identified species was *Echinococcus granulosus sensu stricto* (s.s.). The study results indicate that the disease is still present in several of the investigated regions of Romania, highlighting a potential public health risk.

## 1. Introduction

Cystic echinococcosis is a metacestodosis produced by some species of the genus *Echinococcus*. In Europe, the disease is found in the Mediterranean and Eastern countries [1]. In Romania, the main species is *Echinococcus granulosus sensu stricto* (s.s.). Cystic echinococcosis is found in many species of mammals (e.g., herbivores and omnivores), including humans, which act as intermediate hosts, while definitive hosts are carnivores (e.g., canids, felids, mustelids, and hyaenids). The most common localizations of the hydatid cyst are the liver and lungs, but it can be found in any part of the body

The importance of this disease results from its transmission to humans, sometimes with serious repercussions. In animals, it is usually benign, although it has an important economic impact.

In terms of cystic echinococcosis endemicity, Romania is a mesoendemic country. Persistence in an area or on a farm is related, on the one hand, to the possibility of making trophic contact between intermediate hosts and definitive hosts, and on the other hand, to the control of slaughter and mortality.

*E. granulosus sensu lato* is a complex of different species, including *E. canadensis* (previously known as G6/G7, G8, and G10 genotypes), *E. equinus* (previously known as G4 genotype), *E. felidis*, *E*. *granulosus sensu stricto* (include the genotypes G1, G2, and G3), and *E. ortleppi* [2,3,4,5,6,7,8,9]

The present study provides data on the occurrence and geographical distribution of cystic echinococcosis, and genetic characterization of *Echinococcus granulosus* s.l. from slaughtered cattle in Romania.

## 2. Materials and Methods

In the period 2020–2021, the organs that could be infected from 2693 cattle were examined. The ages of the slaughtered cattle varied between 2 and over 21 years old; they belonged to the Bruna de Maramureș, Bălțată Românească, Aberdeen Angus, Aubrac, Blue Belgique, Charolaise, Siemmental, Holstein, Limousine, Montbeliarde, and mixed breeds. From a total of 2693 slaughtered cattle, 1678 were bred in farms and 1015 in traditional breeding systems, in households.

Out of the total cysts, 10 samples were collected, consisting of proliferating membranes and/or cyst fluid and protoscoleces, which were subjected to molecular biological analyses. The sample selection strategy was based on one sample from each county where positive findings were recorded, except the counties Vâlcea and Mehedinți, where two samples were included (due to the highest number of positive results, Table 1). Extraction of parasitic genomic DNA was performed using the Bioline-ISOLATE II Genomic DNA Kit (BIOLINE, Meridian Bioscience^®^, Tennessee, USA) using the tissue protocol. The technique described by Trachsel et al. (2007) was used with minor modifications [10]. The actual amplification was achieved by a classical PCR, based on the creation of several copies of a sequence of the rrnS gene of size 260–267 bp using primer pairs CEST 3 (5′-YGA YTC TTT TTA GGG GAA GGT GTG-3′) and CEST 5 (5′-GCG GTG TGT ACM TGA GCT AAAC-3′) [10].

For amplification of the PCR products, the My Cycler thermocycler (BioRad^®^, California, USA) was used. The amplicon analysis and control were performed by horizontal electrophoresis in a submersible system, in 1.5% agarose gel, and with the addition of Midori Green fluorescent dye (Nippon Genetics^®^, Europe GmbH, Düren, Germany).

For species confirmation, PCR products were sequenced and compared to those available in GenBank using BLAST alignment (http://blast.NCBI.NLM.nih.gov/blast.cgi (accessed on 29 September 2022).

## 3. Results and Discussion

### 3.1. Descriptive Epidemiology

Of the total 2693 cattle slaughtered, cystic echinococcosis was identified in 66 (2.45%) (Table 1). Although the slaughtered cattle came from 15 counties, cystic echinococcosis was found in 8 counties. Without a plausible explanation, cystic echinococcosis had a higher prevalence (8.69%) in Mehedinti county. Among the positive slaughtered cattle, 32 (1.9%) were from farms and 34 (3.34%) were from traditional breeding systems. The higher prevalence in this last growing system could be due to more frequent trophic contact with the definitive hosts (carnivores), such as canids or felids.

By age categories, cystic echinococcosis was reported in the age categories 2–12 years (6/2341; 0.25%), 13–16 years (11/244; 4.5%), 17–20 years (41/85; 48.23%), and over 21 years (8/23; 34.78%). It can be seen that in the age category 17–20 years, the prevalence is higher. The possibility of reinfections increases with the years as well as the sizes of the cysts. Consequently, the possibility of detection in slaughterhouses is greater. Previous studies have shown that advanced age can influence the presence of cysts [11]. The relatively high prevalence of cystic echinococcosis infection in cattle in Romania can mainly be explained by the movements of dogs (as the main sources for intermediate hosts); it is not perfectly controlled, although it is regulated. There are still dogs without owners. There are still farms where there are free dogs and herds guarded by dogs. The interference of the forest cycle must not be neglected either. There is no clear national control option for this parasitosis. Local initiatives are short-lived and insufficiently extended in time and space.

However, compared to previous years, there is a dramatic reduction in the prevalence of cattle infection in Romania. Thus, in 1997 [12], a prevalence of 40% is reported, and later (2004), Morariu identified a prevalence in cattle of 22.26% (8783/39,272) [13]. In 2014, a study conducted in southern Romania by Mitrea et al. reported a very high rate of cystic echinococcosis infection in cattle (40.1%) (302/754) [14]. Related to this high prevalence in previous years is the very high human incidence rate, 8.63/100,000 inhabitants [15]. However, the results obtained in this study (2.45%) indicate a reduction in the prevalence of infection in cattle. These differences are difficult to explain but could reflect an improvement in the sanitary–veterinary control measures at the farm levels in Romania. In addition, in recent years, improvements in canine population management programs (e.g., microchipping of dogs and their anti-rabies vaccination) have greatly favored the possibility of educating owners (via veterinarians) to deworm their dogs.

In India, a study conducted in 2018 revealed the highest prevalence of cystic echinococcosis (3%) among the species studied (in cattle) [16]. In Brazil, a study conducted over a nine-year period (2009–2017) reported an infection rate of 6.9% [17]. A cross-sectional study carried out in Ethiopia on 610 slaughtered cattle revealed an infection rate of 13.61%, with single or multiple cysts [18].

Table 2 shows the cattle diagnosed with cystic echinococcosis according to breed and gender. The locations of cysts are also specified. The higher number of mixed-breed animals with cystic echinococcosis is explicable because they also constituted the highest number of slaughtered cattle. However, the highest prevalence was found in the Brown breed (8.62%). The higher number of positive females is explained by the comparatively higher number of the slaughtered. The table also shows the predominant locations of cysts in the lungs compared to the liver. Only one case of splenic localization was reported in a female Bălțată Românească breed. For 36 cattle (54.54%), cysts were localized in both the lungs and liver. In one case (1.51%), cysts were localized in the lungs, liver, and spleen. The locations of cysts in cattle, mainly in the lungs (96.96%), compared to the liver (59%), were also revealed in other studies [19]. Thus, bovine lungs are a source of infection for intermediate hosts (carnivores).

The viability of *E. granulosus* cysts in some local cattle breeds is shown in Table 3. Overall, most cysts were either non-fertile (58.29%) or calcified/caseified (38.46%). The fertility rate was very low, regardless of the breed and the organ affected.

A similar study was conducted in Romania by Mitrea et al. (2014). The viability of cysts, although low, was higher in our study (3.23%) than in the previous study (1%). The higher percentage of calcified/caseified cysts may be due to the advanced age of many of the slaughtered cattle [14].

In Ethiopia, in cattle slaughtered in the slaughterhouse, there was a distribution of cysts of 68.67% in the lungs, 14.46% in the liver, 9.64% in both the lungs and the liver, 6.02% in the kidneys, and 1.2% in the heart. Of the 195 cysts checked, 13.85% were fertile, 22.56% were calcified, and 63.59% were non-fertile [18].

A larger study conducted in Bolivia reported a prevalence of 37.5% in cattle slaughtered in the Potosi district and 7.8–16% in the La Paz district, depending on the locality. Molecular studies have shown the presence of *Echinococcus granulosus sensu stricto*, *E. ortleppi* (previously known as genotype G5) [20].

### 3.2. Molecular Epidemiology

Following the PCR analysis of the 10 samples, DNA fragments were identified in 5 samples, all collected from the lungs. The identified species was *Echinococcus granulosus sensu stricto* (previously known as G1 and G3 genotypes) (Table 4).

These results reveal that in Romanian cattle, the zoonotic species *Echinococcus granulosus sensu stricto* is present, which was previously identified, for the first time in Romania, in several hosts, including humans, sheep, and pigs, by Bart et al., (2006) [21]. In addition, in 2014, in southern Romania, *E. granulosus* was identified in cattle, with the two genotypes G1 and G2, presently known as *E. granulosus s.s.* [14].

Recently, in Turkey, Macin et al. (2021) highlighted *Echinococcus granulosus* sensu stricto (G1–G3 genotypes) in both bovine and human cysts, and a human cyst was identified as *E. equinus* (G4 genotype) [22].

Moreover, in most Mediterranean countries, as well as the Balkans, *E. granulosus* s.s. infection predominates, with evolutionary similarities between humans and cattle or other intermediate hosts [1].

## 4. Conclusions

In Romania, in cattle, a prevalence of cystic echinococcosis infection of 2.45% was found.

Older cattle had a higher prevalence. The cysts have been identified mainly in the lungs and liver. Gender and sex appear to be risk factors. Most cysts were non-fertile calcified/caseified.

Based on the PCR analysis, the species *Echinococcus granulosus s.s* were identified. In order to decrease the public health risk, veterinarians should increase their efforts to educate farmers to avoid feeding dogs with infected organs or restrict their free access to cattle feeding and watering areas on farms, where the possibility of shedding the infective forms of the parasite can occur. In addition, regular deworming of dogs can be a valuable prophylactic tool.

## Figures and Tables

**Table 1 vetsci-09-00685-t001:** The prevalence of cystic echinococcosis in different investigated counties of Romania.

No.	County	No. Total Cattle	Positive Cattle (%)
1.	Alba	227	1 (0.44%)
2.	Argeș	233	4 (1.71%)
3.	Arad	12	-
4.	Brașov	127	-
5.	Caraș-Severin	10	-
6.	Dolj	107	-
7.	Gorj	252	6 (2.38%)
8.	Hunedoara	127	6 (4.72%)
9.	Mehedinți	115	10 (8.69%)
10.	Mureș	55	-
11.	Olt	35	-
12.	Sibiu	131	2 (1.52%)
13.	Timiș	62	2 (3.22%)
14.	Teleorman	13	-
15.	Vâlcea	1187	35 (2.94%)
	Total	2693	66 (2.45%)

**Table 2 vetsci-09-00685-t002:** Prevalence and locations of cysts in different breeds of cattle.

No.	Breed	Positive Cattle (%)	Gender (%)		Location of Cysts (%)		
			M	F	Lungs	Liver	Spleen
1	B	20/232 (8.62)	1	19	20 (100%)	9 (45%)	-
2	BR	10/731 (1.36)	1	9	9 (90%)	7 (70%)	1 (11.11)
3	M	35/1204 (2.9)	5	30	34 (97.14)	22 (62.85)	0
4	HF	1/372 (0.26)	0	1	1 (100%)	0	0
Total		66	7 (10.6)	59 (89.4)	64 (96.96)	38 (57.5)	1 (1.51)

Legend: B—Bruna de Maramureș; BR—Bălțată Românească; M—mixed breed; HF—Holstein Friesian; M—male; F—female.

**Table 3 vetsci-09-00685-t003:** Viability of *E. granulosus* cysts in cattle.

Body Region	Breed	Cysts Number	Cysts Viability		
			Fertile No. (%)	Non-Fertile No. (%)	Calcified/Caseified No. (%)
Lungs	B	66	2 (3.03)	40 (60.60)	24 (36.36)
	BR	15	1 (6.66)	8 (53.33)	6 (40)
	M	112	3 (2.67)	65 (58.03)	44 (39.28)
	HF	9	0	4 (44.4)	5 (55.5)
Liver	B	9	1 (11.11)	5 (55.55)	3 (33.33)
	BR	7	0	4 (57.14)	3 (42.85)
	M	35	1 (2.85)	19 (54.28)	15 (42.85)
Spleen	BR	3	0	3 (100)	0
Total		247	8 (3.23)	144 (58.29)	95 (38.46)

Legend: B—Bălțată Românească; BR—Bruna de Maramureș; M—mixed breed, HF—Holstein Friesian.

**Table 4 vetsci-09-00685-t004:** PCR analyzed samples.

No.	Body	Sample Type	Fertility	DNA Identification	Species
1.	lung	protoscoleces	fertile	No	-
2.	lung	proliferative membrane	fertile	Yes	*E. granulosus* s.s.
3.	lung	proliferative membrane	infertile	Yes	*E. granulosus* s.s.
4.	lung	proliferative membrane	infertile	No	
5.	lung	proliferative membrane	infertile	Yes	*E. granulosus* s.s.
6.	lung	proliferative membrane	infertile	Yes	*E. granulosus* s.s.
7.	lung	proliferative membrane + cyst fluid	infertile	No	
8.	lung	proliferative membrane + cyst fluid	infertile	Yes	*E. granulosus* s.s.
9.	liver	proliferative membrane	infertile	No	
10.	liver	proliferative membrane + cyst fluid	infertile	No	

## Data Availability

Data is contained within the article.

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
