# Peer review of "Investigation on Descriptive Epidemiology, Geographical Distribution, and Genotyping of Echinococcus granulosus s.l. in Bovine from Romania"

_vetsci, 2022, doi:10.3390/vetsci9120685_

Round 1

Reviewer 1 Report

1. Compared to previous years,  there is a dramatic reduction in the prevalence of cattle infection in Romania. It suggested the authors to provide the possible resons. 

2. 2693 cattle were involved in this research, what's the sampe size of previous research (Lines 91-93)?

3. 10 samples were chosen for PCR analysis, what's the standard for choosing these samples? The authors should describe this.

4. It suggested the authors provided possible strategies to reduce the echinococosis infection in the conclusions.

5. There are some typos in the references part should be corrected. 

Author Response

Reviewer #1

  1. Compared to previous years, there is a dramatic reduction in the prevalence of cattle infection in Romania. It suggested the authors to provide the possible reasons.

In agreement with the reviewer requirement, the following explanations were added in the revised version of the manuscript “These differences are difficult to explain, but could reflect an improvement of the sanitary-veterinary control measures at the farm levels in Romania. In addition, improvements in the canine population management programs (e.g. microchipping of dogs and their anti-rabies vaccination), in the last years, greatly favored the possibilities of educating owners by veterinarians in order to deworm their dogs.”

  1. 2693 cattle were involved in this research, what's the sample size of previous research (Lines 91-93)?

The requested data for the previously conducted studies in Romania, where they were available, were completed in the revised version of the manuscript resulting in:

- 8783 positive samples from 39272 examined - Morariu (2004)

- 302 positive samples from 754 examined – Mitrea et al. (2004)

  1. 10 samples were chosen for PCR analysis, what's the standard for choosing these samples? The authors should describe this.

The sample selection strategy was based on one sample from each county showing positive results, except the counties Vâlcea and MehedinÈ›i, where two samples were included, since, the registered total number of positive findings were highest compared with other counties.

  1. It suggested the authors provided possible strategies to reduce the echinococosis infection in the conclusions.

According to the reviewer requirement the following sentences were inserted in the Conclusion section of the revised version of the manuscript: “In order to decrease the public health risk, veterinarians should increase their efforts to educate farmers to avoid feeding of dogs with infected organs, or restrict their free access in cattle feeding and watering areas from farms, where the possibility of shedding the infective forms of the parasite can occur. In addition, regularly deworming of dogs can constitute a valuable prophylactic tool.”

  1. There are some typos in the references part should be corrected.

The complete reference list was carefully reviewed and updated according to the MDPI journals reference style.

Reviewer 2 Report

Dear authors,

Congratulations for the communication. I consider that although echinococcosis is suposed to be under control in certain environments, it is still an important parasitism to be considered in several livestock production systems. Moreover, one of the best ways to understand its and other diseases evolution is based on epidemiological studies that should be undertaken by studying a high number of individuals through a more or less extent period time.

Here you have some advice to even improve the communication:

Author Response

Reviewer #2

Dear authors,

Congratulations for the communication. I consider that although echinococcosis is supposed to be

under control in certain environments, it is still an important parasitism to be considered in several

livestock production systems. Moreover, one of the best ways to understand its and other diseases

evolution is based on epidemiological studies that should be undertaken by studying a high

number of individuals through a more or less extent period time.

Dear reviewer, thank you very much for your time and overall positive feedback about the quality of our submission, and your valuable comments helping us to improve the quality of the manuscript.

Here you have some advice to even improve the communication:

Lines 29-30: Please, rewrite the sentence to make it more understandable.

The mentioned sentence was rephrased resulting in “Cystic echinococosis are found in many species of mammals (e.g. herbivores and omnivores), including humans, which act as intermediate hosts, while definitive hosts are carnivores (e.g. canids, felids, mustelids and hyaenids). The most common localization of the hydatid cyst is the liver and lung, but it can be found in any part of the body”

Line 32: benign, although it has an important economic impact.

Corrected according to the reviewer suggestion!

Line 35: or IN a farm

Corrected!

Line 35: …, on the one hand…

Corrected!

Line 36: , and on the other hand, ….

The commas were inserted

Line 40: E. granolusus

Italicized, as requested.

Line 50: bred

Corrected!

Line 53: You refer to Table 2 prior to Table 1. I consider that you should change the order.

Th mention “Table 2” – was deleted from the text.

Line 54: analyses

Corrected!

Line 55: Please, use spaces close to the brackets, use siuperscript for the symbol and add the

commercial trading house and the procedence.

Corrected according to the reviewer requirement!

Line 55: using the tissue protocol. The technique…

Corrected!

Lines 61 and 63. The same comments as for Line 55.

Corrected as the reviewer mentioned!

Line 75: Which carnivores?

The sentence was corrected resulting in “… the definitive hosts (carnivores) such as canids or felids.”

Line 84: Which factors? If dogs are one of the main factors, please, link this concept to the

sentence before.

The sentence was corrected according to the reviewer request!

Line 123: …was higher in our study…

Corrected!

Line 139: Table

Corrected on sentence case!

Line 142: Have you included some results about sheep and carnivores? Where?

No, we did not include any data from sheep or carnivores in the present study! We apologize for the misunderstanding reflected by the old version of the manuscript! We have corrected the text to avoid any confusion resulting in “These results reveal that in Romanian cattle there is present the zoonotic species Echinococcus granulosus sensu stricto, which has been previously identified, for the first time in Romania, in several hosts including humans, sheep and pigs by Bart et al. (2006) [21].”

Line 149: sl

Corrected!

Thank you again for your time, efforts and kindness!